# The Effectiveness of a Diverse COVID-19 Vaccine Portfolio and Its Impact on the Persistence of Positivity and Length of Hospital Stays: The Veneto Region’s Experience

**DOI:** 10.3390/vaccines10010107

**Published:** 2022-01-11

**Authors:** Silvia Cocchio, Federico Zabeo, Giacomo Facchin, Nicolò Piva, Patrizia Furlan, Michele Nicoletti, Mario Saia, Michele Tonon, Michele Mongillo, Francesca Russo, Vincenzo Baldo

**Affiliations:** 1Department of Cardiac Thoracic and Vascular Sciences and Public Health, University of Padua, 35131 Padua, Italy; silvia.cocchio@unipd.it (S.C.); federico.zabeo@unipd.it (F.Z.); giacomo.facchin@studenti.unipd.it (G.F.); nicolo.piva@studenti.unipd.it (N.P.); patrizia.furlan@unipd.it (P.F.); michele.nicoletti@studenti.unipd.it (M.N.); 2Azienda Zero of Veneto Region, 35100 Padua, Italy; mario.saia@azero.veneto.it; 3Regional Directorate of Prevention, Food Safety, Veterinary, Public Health—Regione del Veneto, 30123 Venezia, Italy; michele.tonon@regione.veneto.it (M.T.); michele.mongillo@regione.veneto.it (M.M.); francesca.russo@regione.veneto.it (F.R.)

**Keywords:** COVID-19, SARS-CoV-2, COVID-19 vaccines, vaccine effectiveness, persistence of positivity, length of hospital stay, survival analysis, multivariate regression, Veneto region

## Abstract

The vaccination campaign for the Veneto region (northeastern Italy) started on 27 December 2020. As of early December 2021, 75.1% of the whole Veneto population has been fully vaccinated. Vaccine efficacy has been demonstrated in many clinical trials, but reports on real-world contexts are still necessary. We conducted a retrospective cohort study on 2,233,399 residents in the Veneto region to assess the reduction in the COVID-19 burden, taking different outcomes into consideration. First, we adopted a non-brand-specific approach borrowed from survival analysis to estimate the effectiveness of vaccination against SARS-CoV-2 in preventing infections, hospitalizations, and deaths. We used *t*-tests and multivariate regressions to examine vaccine impact on breakthrough infections, in terms of the persistence of positivity and the length of hospital stays. Evidence emerging from this study suggests that unvaccinated individuals are significantly more likely to become infected, need hospitalization, and are at a higher risk of death from COVID-19 than those given at least one dose of vaccine. Cox models indicate that the effectiveness of full vaccination is 88% against infection, 94% against hospitalization, and 95% against death. Multivariate regressions suggest that vaccination is significantly correlated with a shorter period of positivity and shorter hospital stays, with each step toward completion of the vaccination cycle coinciding with a reduction of 3.3 days in the persistence of positivity and 2.3 days in the length of hospital stay.

## 1. Introduction

The SARS-CoV-2 virus has infected more than 250 million people worldwide, causing over 5 million confirmed deaths [1]. Although COVID-19 carries a higher risk of complications and death for older adults and individuals with comorbidities [2,3,4,5], the involvement of younger sections of the population has also to be taken into account [6,7].

The Veneto region in northern Italy was one of the first areas of Europe to be severely hit by COVID-19 outbreaks [8], and it has reached over 500,000 cases and 11,000 deaths among a population of 4.9 million [9].

Much effort has been made into developing vaccines, with the aim of containing the virus’s transmission, the pressure on critical-care facilities, and the risk of death. Veneto’s vaccination campaign began on 27 December 2020 [10], initially focusing on health workers, resting home residents, vulnerable individuals, and those over 80 years old. It was then gradually extended to other categories and age groups in accordance with the recommendations of the EMA, AIFA, and Italian Ministry of Health [11]. 

At the time of writing, four types of vaccine (mRNA and viral vectors) have been approved for use in Italy: BNT162b2 (Comirnaty, Pfizer–BioNTech, Mainz, Germany), mRNA-1273 (Spikevax, Moderna Biotech, Madrid, Spain), Ad26.COV2.S (Janssen, Janssen-Cilag International NV, Beerse, Belgium), and ChAdOx1 (Vaxzevria, AstraZeneca, Södertälje, Sweden). Completion of the vaccination cycle requires one of the following: two doses of BNT162b2 from 21 to 42 days apart; two doses of mRNA-1273 from 28 to 42 days apart; two doses of ChAdOx1 from 28 to 84 days apart; one dose of Ad26.COV2.S. A positive RT–PCR nasopharyngeal swab 12 months before or more than 14 days after the first injection of any of the approved vaccines also suffices as a complete vaccination cycle [11]. 

As of early December 2021, the following vaccines have been administered in Veneto: BNT162b2 and mRNA-1273 in people 12–60 years old, those over 80, those over 60 with comorbidities, and those explicitly requesting them; a single those of Ad26.COV2.S for people over 60 years old with no specific comorbidities; ChAdOx1 could be used for anyone over 18 without comorbidities up until June 2021, whereas it is now only available as a booster dose for people over 60, who are advised to opt for heterologous vaccination with an mRNA instead. As immunosuppressed individuals have a more limited response to the vaccine, they were given an additional dose of mRNA vaccine at least 28 days after completing the regular vaccination cycle [11]. 

By November 2021, third doses were being administered to people 40+ years old. The EMA also recommended the approval of the extension of the vaccination program also to children between 5 and 11 years old in November 2021 [12]. 

It is crucially important to assess the overall impact of vaccination campaigns on the burden of COVID-19. Randomized, controlled trials (RCTs) are the best-case scenarios for evaluating vaccine efficacy and obtaining data that enable vaccines to be licensed for use. However, the vaccine efficacy seen in trials may not necessarily predict its effectiveness under uncontrolled conditions. For instance, RCTs may not capture the protection afforded by herd immunity resulting from the vaccine’s extensive diffusion. RCTs conducted on groups with certain characteristics may not explain vaccine effectiveness in more diverse populations [13]. RCTs also do not generally consider the use of heterologous vaccination cycles or additional doses. For all these reasons, studies on real-world scenarios are essential [14]. 

Numerous studies of this kind have already shown that vaccines provide a high level of protection against SARS-CoV-2 infection and serious complications of COVID-19, but many of them focused on subgroups of the general population or specific types of vaccine [15,16,17]. These limitations are unavoidable when assessing the impact of vaccination on groups particularly exposed to the risk of infection or trying to ascertain the effectiveness of a particular type of vaccine, but the findings cannot be generalized to a more heterogeneous population. Many studies also suffer from low numerosity of the cohorts analyzed, and that is because monitoring the vaccination status and incidence of the virus in large groups is often unfeasible. 

To overcome these limitations, we examined various aspects of how vaccination as a whole has mitigated the health and social consequences of SARS-CoV-2 in the Veneto region. We first attempted to clarify how vaccination, regardless of the type of vaccine used, has directly affected the rates of infection, hospitalization, and death. This approach has already been used [15,18]. Then, we investigated the correlation between vaccination status and the persistence of positivity and length of hospital stays due to COVID-19. These aspects are particularly important because, while studies on vaccine effectiveness in reducing infection, hospitalization, and death rates clearly demonstrate the success of vaccination, they cannot, in our opinion, fully capture how well the pandemic has been managed as a whole, over time.

## 2. Materials and Methods

Anonymized data on our study population were retrieved from regional databases managed by Azienda Zero (Padua, Italy), a regional institution whose mission consists of ensuring the rationalization, integration, and efficiency of health, social, technical, and administrative services of regional structures. These databases were compiled through a mandatory reporting system, and they contain general information about all the molecular and antigen SARS-CoV-2 tests and the anti-COVID-19 vaccinations performed by the Veneto region, as well as the monitoring of confirmed SARS-CoV-2 cases taken under the charge of the same region.

The latest available updates were 8 October 2021 for molecular and antigen tests and the monitoring of individuals testing positive, and 1 September 2021 for vaccinations. We considered individuals as infected when they tested positive on either an antigen or molecular nasopharyngeal swabs. More than half (55.2%) of these cases had initially been tested with a molecular swab, while the remainder were first identified by a positive antigen swab. More than 86% of the latter cases were confirmed by a positive PCR test within 48 h. When subsequent molecular swabs were negative, the case was recorded as a “false positive”, as suggested by FDA [19]. False-positive tests were then excluded from our analysis.

Our study period extended from 27 December 2020, when the regional vaccine campaign began, and covered 254 days, up to 7 September 2021. For the purposes of our analysis, we only considered individuals who had not tested positive before the starting date of the study and who had at least one swab during the study period. This study population then amounted to 2,233,399 individuals, with 7,019,827 tests (60.1% of which were antigen swabs) taken during the period considered (averaging 3.1 swabs each), and 213,469 individuals testing positive for the virus. 

At any time during the period considered, individuals in our study sample could be unvaccinated, partially vaccinated, or fully vaccinated. There is a well-known delay between inoculation and artificial immunization, and many scientific articles and reports take this into account [20,21,22,23,24]. We considered individuals as partially vaccinated as of 14 days after their first dose of vaccine, and fully immunized as of 7 days after their second or as of 14 days after being vaccinated with the mono-dose Ad26.COV2.S. 

According to our data, some individuals in our sample who never tested positive for SARS-CoV-2 did not appear to have had their second dose of vaccine within a reasonable timeframe. Since subjects who tested positive before the beginning of the vaccination campaign were excluded, we believe that some of these latter cases are due to individuals who refused or deferred their second dose, while some others may be probably due either to missing data or individuals receiving their second dose outside the Veneto region. We then decided to limit the follow-up of partially immunized individuals to the arbitrary threshold of 150 days to take into account potentially belated second doses but, at the same time, not excessively overestimate the effectiveness of the partial immunization.

When necessary, we drew on official public data from the permanent census of the Veneto population, referring to Italian Statistics Institute (ISTAT) records updated to 2020 [25]. 

The data were summarized using percentages and incidence rates, or mean values with their 95% confidence intervals (95% C.I.) as appropriate. Kaplan–Meier curves were used to compare the survival probability among different groups, and log-rank tests were used to identify any significant differences between these curves. A multivariate Cox model was computed, adjusting for covariates such as sex and age, estimating vaccine effectiveness *VE* as
(1)VE=(1−OR)×100
where *OR* stands for the adjusted odds ratio between the risk of an “event” (infection, hospitalization, or death) in immunized and unvaccinated individuals. 

The persistence of individuals’ positivity to the virus was computed as the number of days between their first positive swab and a first negative molecular test performed afterward. Only individuals with such before-and-after swabs were considered when computing the persistence of positivity. Length of hospital stay was computed as the time elapsing between the date of hospitalization and the date of discharge home; patients who died in hospital were excluded from the analysis. When considering the persistence of positivity and length of hospital stay, we compared mean values with *t*-tests, and we conducted multivariate linear regressions with sex, age, and vaccination status as independent variables, and the persistence of positivity or length of hospital stay as the dependent variable.

A *p*-value below 0.05 was always considered statistically significant. 

Data linkage, cleaning, and visualization were performed with specific libraries for data analysis in Python 3.7.0, Kaplan–Meier curves and Cox models were obtained with statistical packages from Python. Linear regressions and *t*-tests were conducted using Excel 2013 and SPSS 27.0. 

## 3. Results

### 3.1. Vaccine Coverage

Judging from data collected by the Veneto Regional Authority, the number of people vaccinated with at least one dose before 1 September 2021 amounted to 3,504,221, and 98% of them were residents of the region. While the proportion of the region’s population that had received at least one dose of vaccine was 70.4%, only 59.2% had been fully vaccinated (Figure 1). 

Several factors contribute to the gap between the numbers of people who were partially vaccinated, as opposed to fully vaccinated. First, anyone given their second dose on schedule after the end of the study period, as well as people considered as fully vaccinated after testing positive for the virus and those receiving one dose of vaccine, would be classified as only partially vaccinated. On the other hand, people who were given a single dose of the Ad26.COV2.S vaccine were classified as fully vaccinated. Approximately 51% of Veneto residents who had been at least partially vaccinated are female, a proportion consistent with the M/F ratio in the region’s general population [25]. 

Vaccine coverage and vaccine brand distributions differed across different age groups (Table 1).

Among those vaccinated with the first dose of BNT162b2, 82.6% had completed their vaccination cycle, almost all of them with a second dose of the same vaccine. Among those inoculated with mRNA-1273 first, 68.1% had received a second dose, almost always (99.8%) with a second dose of the same vaccine. Among those given the first dose of ChAdOx1, only 57.9% had completed their vaccination cycle, with a second dose of the same vaccine in 92.5% of cases. Thus, 0.7% of all the complete vaccination cycles were heterologous, and these cycles mainly involved younger people given the first dose of ChAdOx1, for whom heterologous vaccination became mandatory after 11 June 2021 [26].

### 3.2. Vaccine Effectiveness

#### 3.2.1. Vaccine Effectiveness against Infection

As summarized in Table 2, unvaccinated people had both a higher positivity rate and a higher risk of infection than those who were partially or fully vaccinated. The fully vaccinated were also considerably better protected against infection than the partially vaccinated.

Kaplan–Meier curves and log-rank tests confirmed these findings (Figure 2): at all times, any form of immunization was significantly preferable to no vaccination, and, where comparable, survival probability was significantly lower for the partially vaccinated than for the fully vaccinated population.

The cumulative probability of not contracting the virus decreased at different rates over time in the three subgroups considered, and by 5 months after the second dose, the effectiveness of a full vaccination cycle affords much the same protection as the first dose.

In a Cox multivariate model, vaccine effectiveness against contracting the infection, after adjusting for the effect of sex and age, was 82% (82–83%) for partial vaccination and 88% (87–88%) for full vaccination.

#### 3.2.2. Vaccine Effectiveness against Hospitalization and Death

Table 3 shows the incidence of hospitalizations and deaths for the three subgroups, which were both lower the higher the level of vaccination.

According to the Cox multivariate model, the vaccine effectiveness against hospitalization, after adjusting for sex and age, was 82% (81–84%) for partial vaccination and 94% (94–95%) for full vaccination.

We were not able to distinguish the particular type of hospitalization (intensive care, semi-intensive care, etc.), but for some individuals, the initial clinical status was recorded: we should underline that more than 95% (N = 1405) of those for whom the clinical status is registered as “critical” (N = 1476)—to which those admitted in ICU most likely belong—are unvaccinated.

Vaccine effectiveness against COVID-19-related deaths, after adjusting for sex and age, was 54% (48–60%) for partial vaccination and 95% (94–96%) for full vaccination.

Figure 3 and Figure 4 show the Kaplan–Meier survival curves representing the probability of not being hospitalized and surviving to COVID-19 among unvaccinated, partially vaccinated, and fully vaccinated people: for both outcomes, log-rank tests confirmed that each of the three probability distributions differed significantly from the others.

### 3.3. Impact of Vaccination on the Persistence of Positivity and Length of Hospital Stay

#### Impact of Vaccination on the Persistence of Positivity

As shown in Figure 5, among the people who contracted the virus, those who had been vaccinated remained positive for a shorter period, on average, than those who had not, which means that the former were also likely to recover more quickly than the latter.

On average, positivity seemed to persist for longer the older the individual affected (Figure 6). Our findings also suggest that the time to the first negative molecular swab became shorter with each step toward completion of the vaccination cycle. These results are presented in Table 4, by vaccination status and age group. The results of *t*-tests showed that the unvaccinated, partially vaccinated, and fully vaccinated subpopulations varied considerably in terms of mean persistence of positivity, with all *p*-values lower than 0.001.

The length of hospital stays also correlated strongly with patients’ age (Figure 7).

Table 5 shows the mean length of hospital stays for COVID-19 by vaccination status and age. The results of *t*-tests revealed no significant differences between the unvaccinated and the partially or fully vaccinated groups (*p* = 0.92 and *p* = 0.11, respectively), whereas fully vaccinated patients’ hospital stays were significantly shorter than those of unvaccinated patients (*p* = 0.01).

Table 6 shows the results of multivariate linear regressions were run with the persistence of positivity or length of hospital stay as the dependent variable, and sex, age, and vaccination status as independent covariates. While sex was not significantly associated with the persistence of positivity, this variable showed a linear positive correlation with age and a linear negative correlation with vaccination status. The length of hospital stay showed a significant association with sex, age, and vaccination status: females tended to have shorter hospital stays, older people tended to remain in the hospital for longer, and vaccination status again showed a linear negative association with this variable.

## 4. Discussion

The Veneto region has already reached a good level of vaccination coverage, which will hopefully continue to grow. It is not homogeneous among different age groups because older people were given priority at the start of the vaccination campaign. That said, our findings show that vaccinations are clearly mitigating the effects of COVID-19 regardless of the age group considered. Vaccination has produced favorable results in both clinical and public health terms.

First, we confirmed previous evidence to indicate that SARS-CoV-2 infection poses less of a threat if people are vaccinated. We also found that, even when artificial immunization fails and vaccinated people become infected, they are less prone to serious COVID-19 complications such as hospitalization and death. The risk of both SARS-CoV-2 infection and COVID-19 complications was also found significantly lower in the fully vaccinated than in the partially vaccinated group, confirming the need for a complete vaccination cycle whenever possible. Concerning the risk of infection, Italian Government health policies adopted from 6 August 2021 onwards established that anyone unvaccinated or who had received the first dose of vaccine less than 14 days earlier was obliged to undergo molecular or antigen testing for the virus, in order to access public places (and a negative swab was considered valid for 48 h). This would mean that relatively few positive cases were likely to emerge from the test frequently taken by such people. This could lead to the positivity rate in this subpopulation being underestimated, with a consequent potential underestimation of vaccine effectiveness against infection as well.

It can be inferred that vaccine effectiveness against infection, hospitalization, and death due to COVID-19 decreases over time, especially beyond 150 days after the second inoculation. This is consistent with growing evidence of the efficacy of COVID-19 vaccines waning with time [27]. Given recurrent waves of the pandemic, it is, therefore, fundamentally important to delivering a booster dose of vaccine to those most in need, who were also the people vaccinated first when the vaccination campaign began, so their protection will have faded the most. Some evidence about the association between booster dose and an increase in the IgG titers have already been made [28], and, in our opinion, further research on the efficacy of third doses and on the optimal timing of their administration is now needed.

Our findings suggest a significant reduction in the persistence of positivity in people who have been artificially immunized. The overall average reduction in our sample amounted to 5 days, and according to multivariate regression, each step toward the completion of a vaccination cycle coincide with a reduction of 3.3 days. This aspect is important for two reasons. Reducing the amount of time an infected person remains positive means lowering the overall social costs (less morbidity, fewer school days lost, etc.). It also has a direct influence on the likelihood of a person transmitting the virus to others: Assuming the same social interactions, the longer they remain positive, the greater their chances of infecting someone else. It has also been demonstrated that a shorter period of positivity correlates closely with a lower viral load, meaning that a more persistent positivity could coincide with greater infectivity [29].

Another contribution of our work is the finding that length of hospital stay is also influenced by vaccination status, as it was significantly shorter in our immunized population. The average difference between unvaccinated and fully vaccinated patients’ hospital stays in our sample as a whole was only 2 days, but among people aged 60 or more (the age group most likely to be hospitalized with COVID-19), it ranges from 4 to 5 days. In a multivariate model, each step toward completion of the vaccination cycle coincided with a hospital stay 2.3 days shorter. This is another fundamental issue, as the length of hospital stays strongly influences the burden on healthcare systems. Shorter hospital stays reduce the economic and human resources needed to cope with COVID-19, enabling a faster return to the hospitals’ ordinary activities so heavily affected during the pandemic. One of the aspects to which health authorities have been paying particular attention in assessing the impact of the pandemic (and introducing restrictive measures) is ICU occupancy and ordinary hospitalization rates. Clearly, if further studies confirm that vaccinated people becoming infected and ill enough to need admitting to hospital have shorted stays, then a population with a better vaccination coverage would take longer to overstretch the inpatients care services, and the bar for introducing restrictions on the public’s freedom of movement could be raised.

We are well aware that our work is not without its limitations. For instance, the study was based entirely on information obtained from official regional databases, which meant that the analysis had to be kept at a rather general level. We could not adjust or stratify our analysis on the strength of more specific information, such as the clinical picture or particular risk of individuals. We could not distinguish between types of hospitalization (in intensive care, semi-intensive care, etc.). On other hand, the use of such databases enabled us to conduct statistically significant and robust analyses on large volumes of records. It also gave us a reliable snapshot of the overall impact of vaccination against COVID-19 on the population of the Veneto region as a whole.

It is important to keep in mind that all our findings are influenced by the demographic structure and the vaccine coverage of the Veneto region, as well as by specific viral factors such as the predominant SARS-CoV-2 mutation among our study population. During the beginning of our study period, the distribution of variant of concerns (VOCs) among confirmed SARS-CoV-2 cases in the Veneto region has been quite variable; for instance, in January 2021 at least six different VOCs were circulating, even if the prevailing ones were B.1.177-rel and Alpha (B.1.1.7) e sublineages (Q.x). However, from April 2021, Delta (B.1.617.2) e sublineages (AY.x) started spreading, and from August 2021, almost all the confirmed cases were due to this latter VOC [30]. In the last period, we are witnessing the extremely rapid spreading of the Omicron (B.1.1.529) VOC: since viral variant factors, such as the antigenic mismatch with vaccines or increased transmissibility could affect vaccine effectiveness [31], further studies based on the latest available data are needed.

## 5. Conclusions

In conclusion, part of our work confirms the content of previous reports with a specific focus on experience gained in the Veneto region. Other findings, such as the significant correlations for vaccination status with the persistence of positivity and length of hospital stay, are new, as these issues had yet to be thoroughly investigated. Our work could hopefully serve as a starting point for further research on these variables, which we consider far from secondary aspects of the beneficial consequences of the vaccination campaign.

## Figures and Tables

**Figure 1 vaccines-10-00107-f001:**
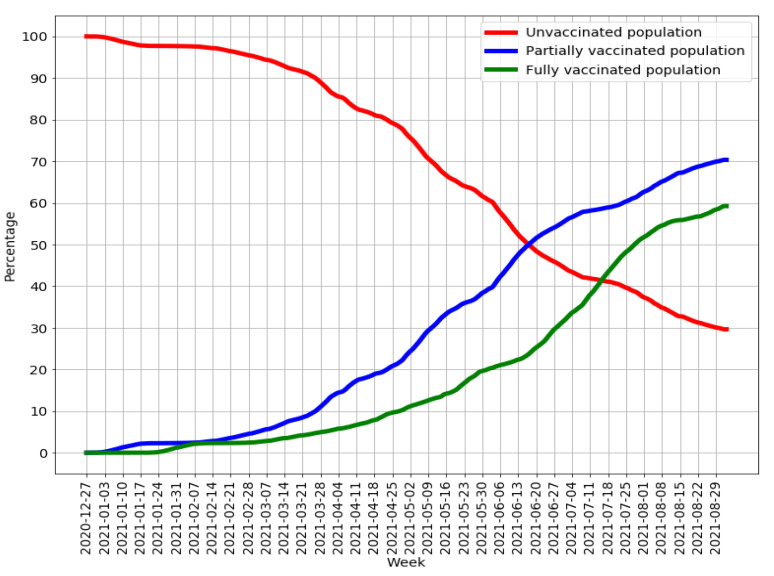
Time trend of the proportions of unvaccinated and vaccinated people in Veneto.

**Figure 2 vaccines-10-00107-f002:**
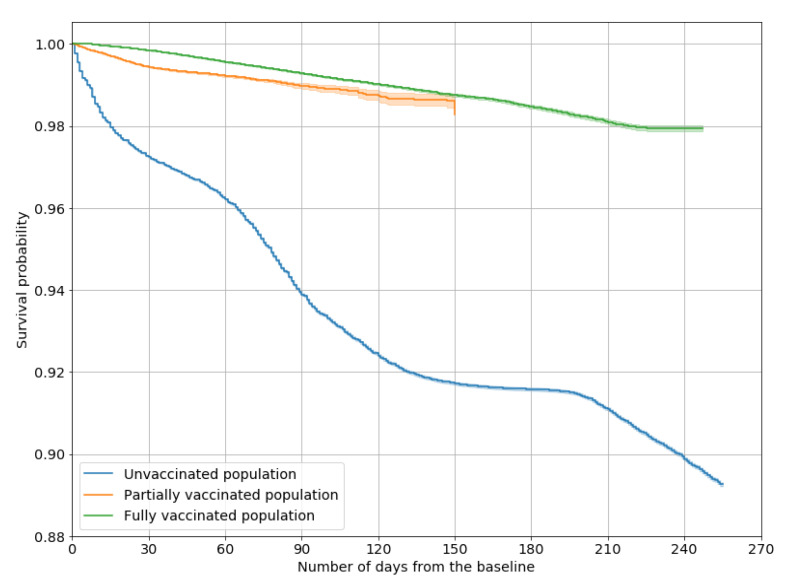
Survival curves representing the cumulative probability of not contracting the SARS-CoV-2 infection, by vaccination status.

**Figure 3 vaccines-10-00107-f003:**
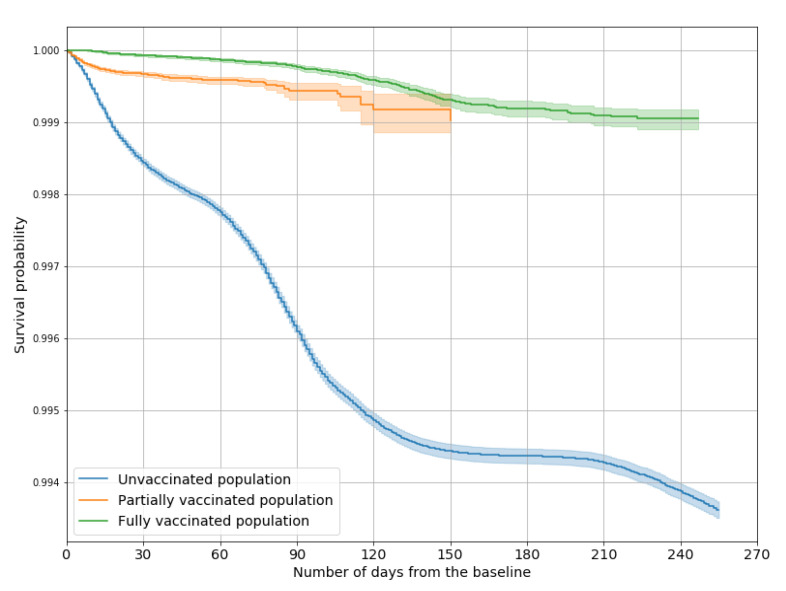
Survival curves representing the probability of not being hospitalized due to COVID-19, by vaccination status.

**Figure 4 vaccines-10-00107-f004:**
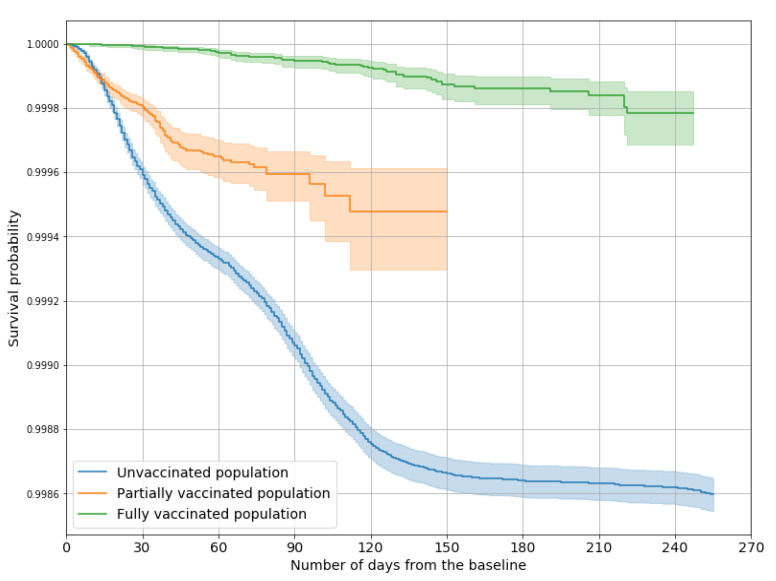
Survival curves representing the probability of surviving from COVID-19, by vaccination status.

**Figure 5 vaccines-10-00107-f005:**
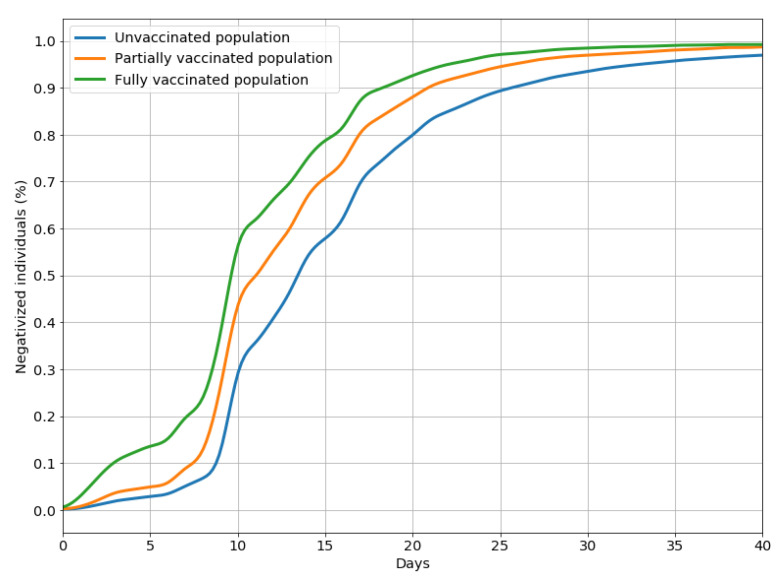
Negativized individuals (subjects tested negative with an RT–PCR molecular swab) with respect to the number of days from the first positive test result, by vaccination status.

**Figure 6 vaccines-10-00107-f006:**
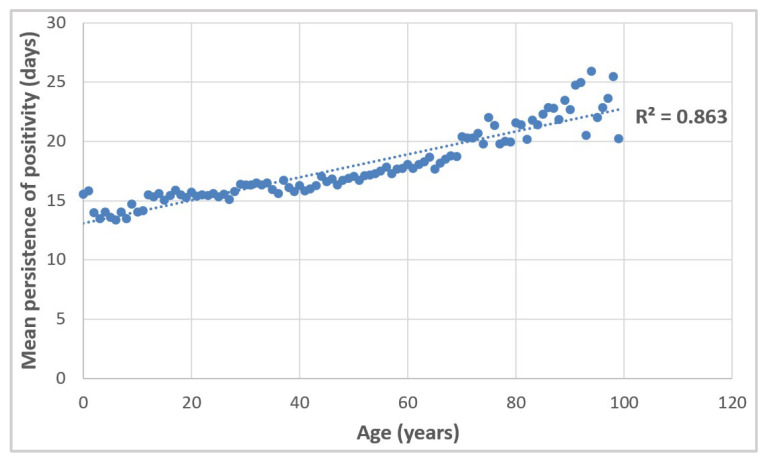
Mean persistence of positivity with respect to the age of the individual affected. Linear trend.

**Figure 7 vaccines-10-00107-f007:**
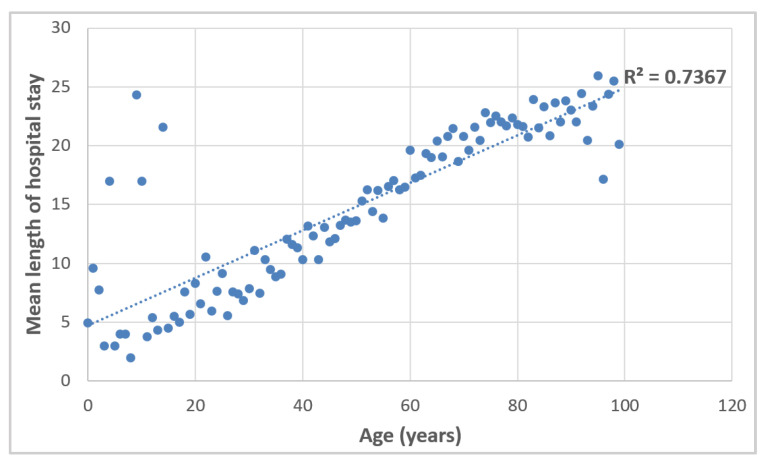
Mean length of hospital stay with respect to patients’ age. Linear trend.

**Table 1 vaccines-10-00107-t001:** Distribution of first doses administered, by vaccine brand and age group.

	Vaccine Brand										
	BNT162b2		ChAdOx1		Mrna-1273		Ad26.COV2.S		All Vaccines		Proportion of the Vaccinated Population
Age Group	N	%	N	%	N	%	N	%	N	%	%
0–11 (N = 493,111)	0	N.D.	0	N.D.	0	N.D.	0	N.D.	0	N.D.	0
12–19 (N = 374,250)	186,726	86.7	178	0.1	28,210	13.1	133	0.1	215,247	100	57.5
20–39 (N = 1,014,067)	545,742	76.7	31,085	4.4	132,046	18.6	1813	0.3	710,686	100	70.1
40–59 (N = 1,541,512)	942,551	81.0	71,693	6.2	142,379	12.2	6443	0.6	1,163,066	100	75.4
60–79 (N = 1,097,653)	489,127	49.3	365,404	36.7	75,098	7.6	63,518	6.4	993,147	100	90.5
80+ (N = 358,540)	279,107	79.6	26,700	7.6	43,319	12.4	1366	0.4	350,492	100	97.8
Total (N = 4,879,133)	2,443,264	71.2	495,060	14.4	421,052	12.3	73,273	2.1	3,432,638	100	70.4

**Table 2 vaccines-10-00107-t002:** Positivity rate and incidence of confirmed cases of SARS-CoV-2 among different groups, by vaccination status.

Age Group	Vaccination Status	Swabs	Person Days	Positives	Positivity Rate (%)	Incidence (× 10,000 Person Days)
0–11	Unvaccinated	440,751	53,388,552	18,935	4.3	3.5
	Partially vaccinated	0	0	0	N.D.	N.D.
	Fully vaccinated	0	0	0	N.D.	N.D.
12–19	Unvaccinated	483,576	51,283,452	21,459	4.4	4.2
	Partially vaccinated	30,645	2,372,904	796	2.6	3.4
	Fully vaccinated	20,167	2,348,846	138	0.7	0.6
20–39	Unvaccinated	1,252,112	118,127,522	50,445	4.0	4.3
	Partially vaccinated	146,325	8,603,820	1964	1.3	2.3
	Fully vaccinated	451,729	18,823,101	1520	0.3	0.8
40–59	Unvaccinated	1,476,304	118,135,226	64,738	4.4	5.5
	Partially vaccinated	213,292	12,327,100	1481	0.7	1.2
	Fully vaccinated	848,114	36,903,807	3215	0.4	0.9
60–79	Unvaccinated	647,178	53,271,459	33,151	5.1	6.2
	Partially vaccinated	115,202	10,062,983	1054	0.9	1.0
	Fully vaccinated	350,392	26,554,328	1973	0.6	0.7
80+	Unvaccinated	235,022	13,486,686	10,945	4.7	8.1
	Partially vaccinated	50,893	2,177,138	635	1.2	2.9
	Fully vaccinated	258,125	13,269,454	1020	0.4	0.8
Total	Unvaccinated	4,534,943	407,692,897	199,673	4.4	4.9
	Partially vaccinated	556,357	35,543,945	5930	1.1	1.7
	Fully vaccinated	1,928,527	97,899,536	7866	0.4	0.8

**Table 3 vaccines-10-00107-t003:** Incidence of hospitalizations and deaths by age groups and vaccination status.

		Hospitalization			Death		
Age Group	Vaccination Status	Number of Cases	Person Days	Incidence (× 100,000 Person Days)	Number of Cases	Person Days	Incidence (× 100,000 Person Days)
0–11	Unvaccinated	74	56,360,445	0.1	1	56,370,577	<0.1
	Partially vaccinated	0	0	N.D.	0	0	N.D.
	Fully vaccinated	0	0	N.D.	0	0	N.D.
12–19	Unvaccinated	39	54,191,996	0.1	0	54,195,717	0
	Partially vaccinated	2	2,650,720	0.1	0	2,651,042	0
	Fully vaccinated	0	2,369,671	0	0	2,369,812	0
20–39	Unvaccinated	591	125,209,790	0.5	4	125,281,911	0
	Partially vaccinated	6	9,722,778	0.1	0	9,730,537	0
	Fully vaccinated	5	19,059,564	0	0	19,060,502	0
40–59	Unvaccinated	2892	127,237,028	2.3	104	127,623,781	0.1
	Partially vaccinated	26	14,420,081	0.2	5	14,475,906	0
	Fully vaccinated	28	37,321,068	0.1	0	37,326,964	0
60–79	Unvaccinated	5233	57,254,697	9.1	942	57,884,010	1.6
	Partially vaccinated	155	11,387,856	1.4	59	11,533,220	0.5
	Fully vaccinated	118	26,994,775	0.4	17	27,034,083	0.1
80+	Unvaccinated	3670	14,418,838	25.5	1824	14,671,443	12.4
	Partially vaccinated	247	2,694,526	9.2	193	2,803,843	6.9
	Fully vaccinated	186	13,504,920	1.4	47	13,535,102	0.3
Overall	Unvaccinated	12,499	434,672,794	2.9	2875	436,027,439	0.7
	Partially vaccinated	436	40,875,961	1.1	257	41,194,548	0.6
	Fully vaccinated	337	99,249,998	0.3	64	99,326,463	0.1

**Table 4 vaccines-10-00107-t004:** Mean persistence of positivity (days) by age group and vaccination status.

	Vaccinal Status		
	Unvaccinated	Partially Vaccinated	Fully Vaccinated
Age Group	Persistence of Positivity (95% C.I.)	Persistence of Positivity (95% C.I.)	Persistence of Positivity (95% C.I.)
0–11	14.1 (13.9–14.3)	N.D.	N.D.
12–19	15.6 (15.4–15.8)	11.8 (11.4–12.1)	10.2 (9.4–11.0)
20–39	16.2 (16.0–16.3)	12.0 (11.7–12.3)	10.6 (10.1–11.0)
40–59	17.2 (17.1–17.4)	13.5 (13.1–13.9)	11.7 (11.3–12.1)
60–79	19.6 (19.3–19.8)	15.8 (15.0–16.6)	12.2 (11.7–12.6)
80+	23.0 (22.5–23.5)	19.2 (18.0–20.4)	13.7 (12.9–14.6)
Tot	17.1 (17.1–17.2)	13.8 (13.5–14.0)	11.8 (11.5–12.0)

**Table 5 vaccines-10-00107-t005:** Mean length of hospital stay (days) by age groups and vaccination status.

	Vaccinal Status		
	Unvaccinated	Partially Vaccinated	Fully Vaccinated
Age Group	Length of Hospital Stay (95% C.I.)	Length of Hospital Stay (95% C.I.)	Length of Hospital Stay (95% C.I.)
0–11	6.5 (4.5–8.4)	N.D.	N.D.
12–19	7.8 (3.9–11.7)	4.0	N.D.
20–39	9.4 (8.6–10.1)	10.4 (5.9–14.9)	5.0 (0–10.5)
40–59	14.6 (14.1–15.1)	17.5 (9.2–25.8)	13.3 (10.0–16.6)
60–79	20.7 (20.1–21.2)	16.2 (13.7–18.7)	16.3 (14.0–18.7)
80+	22.7 (22.0–23.4)	21.5 (18.2–24.8)	18.0 (15.9–20.1)
Tot	18.6 (18.3–18.9)	18.7 (16.7–20.7)	16.7 (15.3–18.2)

**Table 6 vaccines-10-00107-t006:** Summary of multivariate linear regressions.

	Dependent Variable: Persistence of Positivity				Dependent Variable: Length of Hospital Stay			
Model	B	95% C.I.		Significance	B	95% C.I.		Significance
		Lower Limit	Upper Limit			Lower Limit	Upper Limit	
Sex: M = 1, F = 2	0.012	−0.131	0.155	0.868	−1.718	−2.347	−1.088	<0.001
Age	0.088	0.085	0.091	<0.001	0.25	0.232	0.269	<0.001
Vaccinal status: Non vacc. = 0, Part. vacc = 1, Fully vacc. = 2	−3.306	−3.493	−3.119	<0.001	−2.317	−3.246	−1.389	<0.001

## Data Availability

The data supporting the findings of this study are available from the corresponding author upon reasonable request.

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
