# Peer review of "The Effectiveness of a Diverse COVID-19 Vaccine Portfolio and Its Impact on the Persistence of Positivity and Length of Hospital Stays: The Veneto Region’s Experience"

_vaccines, 2022, doi:10.3390/vaccines10010107_

Round 1
Reviewer 1 Report
Thank you for allowing me to review your manuscript.
The Abstract, Introduction, Materials and Methods, Results, Discussion and References sections contain the appropriate components.
There are easily correctable English grammar/neologism issues (examples include: lines 62, 81, and 229 - this is not meant as a comprehensive list).
Figure 2 needs to be moved away from the preceeding text as it obscures the text in the copy provided to me.
You do not mention the potential impact and implications of the variants of concern (VOCs) on your findings. Acknowledgement and commentary on this important aspect is essential as it was/is relevant during the time of this work.
Are you able to exclude other variables which may have impacted hospital length of stay? Many health care systems changed criteria for hospital admission and continued hospital stay during 2021 which led to decreased hospital length of stay prior to widespread vaccination.
Author Response
1) There are easily correctable English grammar/neologism issues (examples include: lines 62, 81, and 229 - this is not meant as a comprehensive list).
We apologize for our misprints or English grammar issues: we read again our manuscript and we tried to solve both noticed and unnoticed issues. We hope that the manuscript is now ok. We were not aware that the word “negativized” was a neologism, as the reviewer kindly pointed out: we maintained this word on the Y-axis of Figure 5 just for the sake of brevity, whereas we widely explained the precise meaning of that word in the legend of the same Figure.
2) Figure 2 needs to be moved away from the preceeding text as it obscures the text in the copy provided to me.
We fixed this inconvenience.
3) You do not mention the potential impact and implications of the variants of concern (VOCs) on your findings. Acknowledgement and commentary on this important aspect is essential as it was/is relevant during the time of this work.
Unfortunately, the specific SARS-CoV-2 mutation detected among confirmed cases was not registered in our data, so that we were not able to consider this variable in our study. However, we totally agree with the reviewer that such viral factors could widely affect vaccine effectiveness. In this respect, we added a few lines on the “Discussion” to summarize the VOCs distribution which took place in the Veneto region during the study period, and we also underlined the importance of further research to assess the vaccine effectiveness in a context with different prevailing VOCs.
4) Are you able to exclude other variables which may have impacted hospital length of stay? Many health care systems changed criteria for hospital admission and continued hospital stay during 2021 which led to decreased hospital length of stay prior to widespread vaccination.
We agree with the reviewer that such changes could significantly bias both the number of hospitalizations and the length of hospital stay. However, according to our knowledge, the Veneto Region did not implement any changes on the criteria for hospital admission or on the continued hospital stay during our study period, and for this reason, we did not account for such a variable when studying such indicators. Moreover, our choice to let the study begin exactly when the vaccination campaign began has been precisely made to smooth this kind of “inhomogeneity”: potential new policies or renewed criteria would have necessary impacted both on vaccinated and unvaccinated subgroups.
Reviewer 2 Report
The topic is of obvious importance and I agree with the authors that real world data are needed to understand the true effect of vaccination against Sars-CoV-2. The study is in general well written, besides a few parts. The conclusions are supported by the results, especially with the weaknesses addressed.
Specific comments:
" A positive RT-PCR nasopharyngeal swab 12 months before or more than 14 days after a first injection of any of the approved vaccines also suffices as a complete vaccination cycle." - provide a reference or policy for this.
"Vaccination has produced positive results in both clinical and public health terms." - change "positive" to "favorable".
"This is consistent with growing evidence of the efficacy of COVID-19 waning with time [27]." - do you mean the efficacy of vaccines waning?
" so heavily penalized during the pandemic." - change to "affected"
Author Response
1) " A positive RT-PCR nasopharyngeal swab 12 months before or more than 14 days after a first injection of any of the approved vaccines also suffices as a complete vaccination cycle." - provide a reference or policy for this.
We now added – in correspondence to the above-cited sentence - another reference to [11], which is a repository of all the official indications on vaccinations issued by the Italian Ministry of Health. Particularly, the specific policies the reviewer is asking for can be found in the documents issued on 21 July 2021 and 9 September 2021.
2) "Vaccination has produced positive results in both clinical and public health terms." - change "positive" to "favorable".
We modified the sentence following the reviewer’s suggestion.
3) "This is consistent with growing evidence of the efficacy of COVID-19 waning with time [27]." - do you mean the efficacy of vaccines waning?
We guess it is a misprint, and we correct the sentence, in that we are clearly referring to the “efficacy of vaccines waning”, as the reviewer pointed out.
3) " so heavily penalized during the pandemic." - change to "affected"
We modified the sentence following the reviewer’s suggestion.
Reviewer 3 Report
In this paper, the authors present effectiveness data from the Veneto region of North-East Italy, from a cohort study among over 2 million residents. They show that vaccination status is significantly associated with greater protection against disease, in terms of infection rates, hospitalization rates, length of stay and duration of positivity. The findings confirm that even among those receiving one dose of a vaccine, protective effects are visible, and recipients of a full vaccination schedule are even more protected.
In the current climate of recurrent pandemic waves and persistent anti-vaccine sentiment among certain segments of the population, I think that these findings are important, and deserve to be brought to the attention of the community to underline that our efforts to achieve high vaccination rates are paying off, and worth the continued struggle.
I have some minor comments for the authors’ consideration.
- Could the authors explain in greater detail what databases they used exactly? It currently says “regional databases”, but it is not clear from what – health insurance claims? Mandatory reporting systems? Some details of what the database’s original purpose was, and who managed it, would be helpful.
- Throughout the paper, there are many percentages where the decimal is represented by a comma – this should be corrected to a period (including on the figures).
- Line 62, there appears to be a word missing regarding the administration of an additional dose of mRNA in immunosuppressed individuals – I presume you mean at least 28 days AFTER the regular vaccination cycle?
- Line 171, there are a few mistakes here – firstly, it should read “except FOR isolated cases”, but if these are isolated, how can they come to 100%, as indicated in the parentheses? (Note also, this should read “approximately” and not “approximatively”).
- Figure 2, the legend on the Y-axis reads “survival probability”, but these are not curves of the cumulative risk of survival (or death), but rather the probability of contracting the virus. According to the legend of the figure, this is the cumulative probability of contracting the virus over time, but that would mean that you are less likely to get COVID as time goes by – whereas the reverse is true. Immunity wanes with elapsing time since vaccination, and therefore the probability shown on the graph is probably the infection-free survival. This should be clarified and the legend corrected accordingly. The same applies to Figure 3, where I suspect the authors mean “hospitalization-free survival”. Conversely, Figure 4 is correct
Author Response
1) Could the authors explain in greater detail what databases they used exactly? It currently says “regional databases”, but it is not clear from what – health insurance claims? Mandatory reporting systems? Some details of what the database’s original purpose was, and who managed it, would be helpful.
We added a few lines on Material and Methods to give greater detail of our databases.
2) Throughout the paper, there are many percentages where the decimal is represented by a comma – this should be corrected to a period (including on the figures).
We apologize for this inconvenience: we now fixed our manuscript, and now hopefully each period represents a decimal separator whereas each comma stands for a thousand separators. We also added the first decimal number on some percentages of Table 1 after noticing that in the provided version of the manuscript they have been accidentally omitted.
3) Line 62, there appears to be a word missing regarding the administration of an additional dose of mRNA in immunosuppressed individuals – I presume you mean at least 28 days AFTER the regular vaccination cycle?
We apologize for the misprint: we corrected the sentence.
4) Line 171, there are a few mistakes here – firstly, it should read “except FOR isolated cases”, but if these are isolated, how can they come to 100%, as indicated in the parentheses? (Note also, this should read “approximately” and not “approximatively”).
We reformulated the sentence in a clearer way and we corrected the misprint.
5) Figure 2, the legend on the Y-axis reads “survival probability”, but these are not curves of the cumulative risk of survival (or death), but rather the probability of contracting the virus. According to the legend of the figure, this is the cumulative probability of contracting the virus over time, but that would mean that you are less likely to get COVID as time goes by – whereas the reverse is true. Immunity wanes with elapsing time since vaccination, and therefore the probability shown on the graph is probably the infection-free survival. This should be clarified and the legend corrected accordingly. The same applies to Figure 3, where I suspect the authors mean “hospitalization-free survival”. Conversely, Figure 4 is correct
We apologize for the misunderstanding: we probably used ambiguous terminology. For each of the three “events” (infection, hospitalization and death) we intended “survival probability” as the (cumulative) probability of having not experienced the “event”. For instance, the x-axis in Figure 2 represents the time (days) elapsed since the beginning of the follow up whereas the y-axis provides the probability of having not yet contracted the infection in correspondence to the considered x value. We tried to make it more clear in the legend of the Figures.